# Self-Medication with Antibiotics for Protection against COVID-19: The Role of Psychological Distress, Knowledge of, and Experiences with Antibiotics

**DOI:** 10.3390/antibiotics10030232

**Published:** 2021-02-25

**Authors:** Airong Zhang, Elizabeth V. Hobman, Paul De Barro, Asaesja Young, David J. Carter, Mitchell Byrne

**Affiliations:** 1Health & Biosecurity, Commonwealth Scientific and Industrial Research Organisation, Brisbane, QLD 4102, Australia; Paul.Debarro@csiro.au (P.D.B.); Asaesja.Young@csiro.au (A.Y.); 2Land & Water, Commonwealth Scientific and Industrial Research Organisation, Brisbane, QLD 4102, Australia; Elizabeth.V.Hobman@csiro.au; 3Faculty of Law, University of Technology Sydney, Sydney, NSW 2007, Australia; David.Carter@uts.edu.au; 4College of Health and Human Sciences, Charles Darwin University, Darwin, NT 0815, Australia; mitchell.byrne@cdu.edu.au

**Keywords:** knowledge of antibiotics, antimicrobial resistance, antibiotic use for prevention, perceived health risk, coronavirus pandemic, psychological distress

## Abstract

Self-medication with antibiotics is a major contributing factor to antimicrobial resistance. Prior research examining factors associated with antibiotic self-medication has focused on an individual’s knowledge about antibiotics, antibiotic usage practices, accessibility to antibiotic medication, and demographic characteristics. The role of psychological distress associated with perceived health risks in explaining antibiotic self-medication is less understood. This study was designed to address this knowledge gap in the context of the COVID-19 pandemic in Australia. An online survey of 2217 participants was conducted at the height of the initial outbreak and revealed that 19.5% of participants took antibiotics to protect themselves from COVID-19. Multivariate logistic analysis examined the predictors of taking antibiotics for protection against COVID-19. An integrative framework developed from the results illustrates potential pathways and facilitating factors that may contribute to prophylactic self-medication with antibiotics. Specifically, COVID-19 pandemic-induced psychological distress was significantly positively related to self-medication. Preventive use of antibiotics was also facilitated by a lack of understanding about antibiotics, inappropriate antibiotics usage practices, the nature of the patient-doctor relationship, and demographic characteristics. The findings highlight that to combat antimicrobial resistance due to self-medication, interventions need to focus on interrupting entrenched behavioural responses and addressing emotional responses to perceived health risks.

## 1. Introduction

### 1.1. Antimicrobial Resistance and Self-Medication with Antibiotics

Antimicrobial resistance (AMR) is one of the greatest threats facing humanity. Drug-resistant infections cause at least 700,000 deaths globally each year. Without intervention, this mortality rate has been projected to reach 10 million deaths annually by 2050 [1]. Self-medication with antibiotics has been identified as a significant contributing factor to AMR [2,3,4,5,6]. Self-medication with antibiotics is defined as an inappropriate and irrational antimicrobial use, whereby individuals treat self-diagnosed symptoms/illnesses without prescriptions, medical advice, and supervision [4,7]. Research has shown that self-medication with antibiotics is widely practised, even in developed countries with advanced healthcare systems and strict regulations [4,8,9,10,11,12,13].

Tackling AMR has become an urgent priority for policy makers globally [14]. Research so far has suggested that self-medication with antibiotics is often caused by a lack of knowledge about antibiotics, widespread availability of antibiotic medications due to absence of government regulations and enforcement, and incorrect practices such as not completing the full treatment course and keeping the remaining antibiotic medications [15,16,17,18]. These insights have informed the design of campaign strategies globally in managing antibiotic misuse, wherein the focus is on education and dissemination of knowledge [19,20,21,22]. However, given the context in which antibiotic self-medication occurs—that is, individuals acting to protect their health—it is surprising that the role of psychological distress in self-medication with antibiotics is yet to be examined. Moreover, self-medication with antibiotics for the purpose of illness prevention is less understood. The COVID-19 pandemic offers an ideal opportunity to examine these issues and bridge our knowledge gaps.

### 1.2. COVID-19 Pandemic and Psychological Distress

The number of Coronavirus cases in Australia began to significantly increase in early March 2020, coinciding with the World Health Organisation declaring COVID-19 a global pandemic on the 11 March 2020. Public concern was evidenced by panic buying of essential products with fear fuelled by media images of emptied supermarket shelves and daily COVID-19 news reports in Australia and worldwide. Australian federal, state, and territory governments implemented multiple measures, including restrictions on international and domestic travel, the closure of borders between states, and ring-fencing cities to contain the movement of people.

Reporting of, and response to, the COVID-19 pandemic instilled a strong sense of an out-of-control crisis. Many studies have detailed the significant impact on mental health among the general public, reflected in rising stress, fear, anxiety, and depression [23,24,25]. Consistent with public health advice, the general community took measures to protect themselves from COVID-19, such as wearing face masks and isolating from the community at large. However, extreme preventative measures were also reported, such as intentionally inhaling or ingesting household cleaning products with the intent of preventing COVID-19 transmission [26]. The current study sought to examine measures taken by the public in Australia to further protect themselves from contracting COVID-19 with respect to the use of antibiotics.

### 1.3. Psychological Distress and Self-Medication

It is not uncommon for people experiencing psychological distress to practise self-medication as a management strategy, including the use of substances such as alcohol and other drugs [27,28,29]. Self-medication may increase a person’s belief that they can cope with stress and anxiety [27,28,30]. The role of psychological distress induced by a potential infection in self-medication with antibiotics is yet to be examined. Research on self-medication with antibiotics has largely focused on factors at the individual and healthcare professional levels [11,31,32]. At an individual level, self-medication is positively related to storing antibiotics at home and having easy access to antibiotics [33,34]. The relationship between other individual factors such as knowledge of antibiotics, age, gender and education, and self-medication using antibiotics has yielded conflicting data, with both supporting and opposing findings reported ([10], for a review, see [11]). At the healthcare professional level, a key contributor is the acquiescence of doctors to pressure from patients for an antibiotic prescription [11].

### 1.4. The Present Study

While it has been demonstrated that COVID-19 has significantly impacted people’s mental health [23,24,25], what measures people are taking to manage such psychological distress is less well understood. Examining this research question in the context of inappropriate use of antibiotics and AMR, we aimed to investigate the role of psychological distress in self-medicating with antibiotics. Hence, this study explored whether the psychological distress induced by the COVID-19 pandemic is associated with preventive use of antibiotics (i.e., as a protective measure against contracting COVID-19). In particular, we examined the relationship between psychological distress associated with COVID-19 and self-medication with antibiotics, along with other known contributing factors, including knowledge about antibiotics, experience in using antibiotics, the nature of the patient relationship with doctors, and demographic characteristics. At the height of the initial COVID-19 outbreak between 16 March to 1 April 2020 in Australia, an online survey was conducted with an age and gender representative sample of the general public from six major capital cities. Multivariate logistic regression analysis was conducted to examine predictors of self-medication with antibiotics for protection against COVID-19. An integrative model was then constructed to illustrate the pathways and facilitating factors for self-medication with antibiotics in the context of the COVID-19 pandemic.

## 2. Results

Table 1 presents the demographic characteristics of the sample of participants. 2790 potential participants from six major capital cities were invited to participate. Among this sample, 2481 started the survey, but only 2217 completed it. The completed-to-viewed rate is therefore 79.5%. The participants from the six major capital cities were age and gender representative of each city.

### 2.1. Measures Taken against COVID-19

Of the 2217 participants, 432 (19.5%) reported that they were taking antibiotics to protect themselves against the coronavirus at the time of survey. Table 2 presents the comparison between the two groups—those who took antibiotics to protect themselves from the coronavirus and those who did not—on the measures they took against COVID-19. These two groups are referred to as antibiotic takers and antibiotic non-takers or the non-takers. Due to the unequal sample sizes of the two groups, the comparison analysis was conducted with equal variances not assumed.

As shown in Table 2, most participants (~80%) had taken protective measures against being infected with the Coronavirus, except for wearing a mask. Comparisons between the antibiotic takers and non-takers revealed that antibiotic takers were more cautious than the non-takers, especially in wearing a mask (63.2% vs. 18.2%; *χ*^2^[1, *n* = 2217] = 358.64, *p* < 0.001).

### 2.2. Predictors of Taking Antibiotics for Protection against COVID-19

To examine what variables are associated with taking antibiotics for protection against COVID-19, multivariate logistic regression analysis was applied, with demographic characteristics, COVID-19 associated psychological distress, knowledge of antibiotics, experiences in using antibiotics, and relationship with doctors as additional explanatory variables. Table 3 presents the percentages of categorical variables, means, and standard deviations of continuous variables, and the odds ratios (OR) of each variable in predicting taking antibiotics for protection against COVID-19. Further, 2-tailed *χ*^2^ tests or 2-tailed *t*-tests were conducted to examine the differences on those significant variables between antibiotic takers and antibiotic non-takers to develop more nuanced understanding. The following sections provided more detailed description of the results from these analyses.

#### 2.2.1. Demographic Characteristics

As shown in Table 3, age (*OR* = 0.89, *p* = 0.010), gender (*OR* = 1.29, *p* = 0.048), and education (*OR* = 1.13, *p* = 0.001) were all significantly associated with taking antibiotics for COVID-19 protection. Results of *t*-tests show that antibiotic takers were significantly younger (M = 2.74, SD = 1.37) than the non-takers (M = 3.70, SD = 1.64; *t*(2215) = −11.24, *p* < 0.001), and were more educated (M = 4.37, SD = 1.73) than the non-takers (M = 3.78, SD = 1.82; *t*(2215) = 6.12, *p* < 0.001). There were significantly more male participants among the antibiotic takers group (56.5%) than among the non-takers (48.6%), *χ*^2^(1, *n* = 2217) = 8.58, *p* = 0.003.

Moreover, participants working in a health profession was 1.64 times more likely than those not working in a health profession to take antibiotics (*OR* = 1.63, *p* = 0.007). Compared to antibiotic non-takers, a higher proportion of antibiotic takers were trained, or worked, in a health-related profession (27.5% vs. 10.8%; *χ*^2^[1, *n* = 2217] = 81.30, *p* < 0.001). However, having a family member working in health profession was not related to antibiotics taking (*OR* = 1.13, *p* = 0.451).

#### 2.2.2. Psychological Distress over COVID-19 Pandemic

COVID-19 pandemic induced psychological distress was positively associated with taking antibiotics for COVID-19 (*OR* = 1.35, *p* < 0.001). Antibiotic takers reported significantly higher level of psychological distress (M = 3.41, SD = 1.04) than the non-takers (M = 2.85, SD = 0.97; *t*(2215) = −10.44, *p* < 0.001; 1 = not at all, 3 = moderately, 5 = very much). Further analyses were conducted to compare the two groups on each of the emotion items that composed the measurement of psychological distress. As shown in Table 4, the emotional reactions reported by antibiotic takers were all above the mid-point of the scale, and the non-takers’ emotional reaction was mostly below the mid-point. The strongest contrasts are that the antibiotic takers felt more panicked (M = 3.26 vs. M = 2.30) and scared (M = 3.45 vs. M = 2.79).

#### 2.2.3. Knowledge about Antibiotics

There was misunderstanding among some participants that antibiotics can treat the common cold (18.9% stated Yes to this statement) and viral infection (28.1% stated Yes to this statement). However, such misunderstandings were not associated with taking antibiotics. Misunderstanding that antibiotics could treat the flu, and that antibiotics could not treat bacterial infections, were both significantly associated with taking antibiotics for COVID-19 (*OR* = 1.60, *p* = 0.001; *OR* = 1.99, *p* < 0.001; respectively). Results from *χ*^2^ tests revealed that significantly more antibiotic takers (50.7%) than non-takers (30.1%) believed that antibiotics can treat the flu (*χ*^2^[1, *n* = 2217] = 65.75, *p* < 0.001). Most antibiotic non-takers (79.5%) correctly knew that antibiotics can treat bacterial infections, while only 53.2% of antibiotic takers held this understanding (*χ*^2^[1, *n* = 2217] = 125.82, *p* < 0.001).

Surprisingly, self-reported knowledge on the difference between viral and bacterial infections was positively associated with taking antibiotics (*OR* = 1.16, *p* = 0.028). On average, antibiotic takers believed that they had more knowledge (M = 3.20, SD = 1.06) than the non-takers (M = 2.87, SD = 1.01; *t*(2215) = 6.07, *p* < 0.001; 1 = know nothing at all, 5 = know a lot). In addition, 74.4% of the participants had heard about the term of antibiotic resistance. However, whether participants had heard about antibiotic resistance or not was not related to taking antibiotics for COVID-19 (*OR* = 0.94, *p* = 0.811).

#### 2.2.4. Experience in Using Antibiotics

A substantial proportion of participants (35.6%) reported that they had taken antibiotics for cold and flu. But such behaviour did not predict taking antibiotics for COVID-19 (*OR* = 1.35, *p* = 0.062). When treated with antibiotics, 35.1% of the participants reported that they had stopped taking the prescribed antibiotics when they started feeling better. However, this was not associated with taking antibiotics for COVID-19 (*OR* = 0.79, *p* = 0.134).

Both taking leftover antibiotics from previous prescriptions or friends’ or family members’ prescriptions and reporting that they could easily acquire antibiotics from family/friends’ household significantly predicted taking antibiotics for COVID-19 (*OR* = 1.85, *p* < 0.001; *OR* = 2.03, *p* < 0.001; respectively). Results from χ^2^ tests indicated that significantly more antibiotic takers than the non-takers had taken antibiotics that were left over (48.1% vs. 17.1%; *χ*^2^[1, *n* = 2217] = 212.42, *p* < 0.001); and significantly more antibiotic takers than the non-takers found they could easily get antibiotics from their family and/or friends (43.1% vs. 11.8%; *χ*^2^[1, *n* = 2217] = 256.03, *p* < 0.001).

The results also indicated that the more frequently participants took antibiotics in the past 12 months, the more likely they were to have taken antibiotics to protect themselves against COVID-19 (*OR* = 1.24, *p* < 0.001). Results of *χ*^2^ tests revealed that antibiotic takers took antibiotics more often than the non-takers in the past 12 months (*χ*^2^[4, *n* = 2217] = 149.90, *p* < 0.001; see Table 5).

#### 2.2.5. Relationship with Doctors

Nearly half of the participants (47.1%) reported that it was easy to get antibiotics from a doctor. However, this was not associated with taking antibiotics for COVID-19. Among the participants, almost one-third (32.2%) had asked his/her doctor to prescribe antibiotics for an illness, and this request was significantly related to taking antibiotics for COVID-19 (*OR* = 1.36, *p* = 0.048). Results from the *χ*^2^ test revealed that significantly more antibiotic takers (49.8%) than the non-takers (28.0%) had asked their doctors to prescribe antibiotics for an illness (*χ*^2^[1, *n* = 2217] = 110.78, *p* < 0.001).

Participants reported high level of trust in their doctor’s decision on whether they need antibiotics (M = 3.96, SD = 1.16), and stated they would follow their doctor’s instruction on how to use antibiotics (M = 4.18, SD = 1.03) (both were measured on a 5-point scale). However, these two variables were not associated with taking antibiotics for COVID-19 (*OR* = 0.93, *p* = 0.295; *OR* = 0.90, *p* = 0.188; respectively).

## 3. Discussion

In a sample of the general population of Australia, we found that taking antibiotics as a preventative measure against COVID-19 was quite widespread. In fact, almost 1 in 5 (19.5%) took antibiotics for protection against COVID-19. We then examined a range of potential drivers of self-medicating with antibiotics in the context of the COVID-19 pandemic. To assist in the synthesis of our results, we have constructed an integrated framework to summarise the main findings of our study (see Figure 1). This framework illustrates the pathways and facilitating factors that are associated with taking antibiotics for protection against COVID-19.

As the framework suggests, there were several factors associated with taking antibiotics for prevention against COVID-19. We found that participants who took antibiotics for COVID-19 reported significantly higher level of psychological distress as compared to those who did not. This result is consistent with findings widely reported since the outbreak of COVID-19 [23,24,25]. We observed that antibiotic takers, as compared to antibiotic non-takers, were significantly more panicked and scared about the potential infection. They also felt more helpless, angry, worried, and sad. In addition to reporting higher levels of psychological distress linked to COVID-19, antibiotic takers were also more cautious in their preventative practices than antibiotic non-takers. Indeed, antibiotic takers were more likely to take a range of additional measures to protect themselves against COVID-19, such as wearing a mask and avoiding social interactions. It is possible then that taking antibiotics may be an extra measure that these participants took, presumably to give them a sense of control in managing the potential threat of contracting the illness.

Several other factors were also associated with taking antibiotics for preventative purposes and were largely consistent with those that have been identified as key determinants of self-medication with antibiotics in previous studies [11,15,31,32,34,35]. First, a lack of knowledge about antibiotics was shown to be quite common among the Australian public, and in some instances, this lack of knowledge was significantly associated with taking antibiotics for COVID-19. Here we found that a substantial proportion of participants misunderstood that antibiotics could treat the common cold (18.9%) and viral infections (28.1%). These rates of misunderstanding however are much lower than most of the European countries (28% and 48%, respectively) as of 2018—one year prior to the current study [17]. The rates are also lower compared to low- to middle-income countries such as Thailand (52.3% and 49.8%, respectively) [36].

We observed even higher levels of misunderstanding on other aspects—that is, the fact that antibiotics could treat the flu (34.1% agreed with this incorrect fact) and not knowing that antibiotics can treat bacterial infections (25.6% disagreed with this correct fact). Importantly, misunderstanding either of these two facts was significantly associated with taking antibiotics for COVID-19, increasing the odds by 1.60 to 1.99, respectively. Approximately half of the antibiotic takers did not hold correct knowledge of these two facts. Interestingly though, antibiotic takers tended to self-report that they knew more about the difference between viral and bacterial infection (as compared to antibiotic non-takers), despite holding less knowledge of the role of antibiotics in treating these types of infections. This corollary suggests that there may be some benefit in dispelling certain myths surrounding the correct use of antibiotics.

Second, from the aspect of experience with, or usage of, antibiotics, we found that taking leftover antibiotics was considerably prevalent in our sample (23.2% reported partaking in this practice), and a significant number of participants (17.9%) reported that it was easy for them to obtain antibiotics from friends or family members. Having taken leftover antibiotics in the past or reporting that it was easy for them to obtain antibiotics from friends and family members was strongly associated with taking antibiotics for COVID-19. While antibiotic dispensation without a prescription through community pharmacies is a common practice, especially in low- and middle-income countries (for a review, see [16]), it is noted that antibiotics can only be acquired with doctor’s prescriptions in Australia. This regulation in Australia may reduce the extent to which antibiotic medication is overused. However, our findings suggest that access to antibiotics may still be relatively easy via keeping leftovers or obtaining antibiotics from friends and family members.

The prevalence of self-medication with leftover antibiotics for COVID-19 is in line with previous research that has revealed that keeping leftover antibiotics is linked to self-medication with antibiotics in general [8,12,31,32,33,34]. The use of leftover antibiotics reflects poor adherence to the treatment protocol as recommended by the administering doctor. We also found that antibiotic takers tended to use antibiotics more often than those who did not. However, the cause for (previous) use of antibiotics and how the medication was acquired is not clear and is beyond the scope of the present study. Nevertheless, this result suggests that taking antibiotics could be an entrenched behavioural response to any perceived health risks, including COVID-19 as in the current situation.

Third, our study revealed that nearly half of the participants (47.1%) found that it was easy to obtain antibiotics from their doctors and a large percentage of participants (32.2%) had previously asked for antibiotics from their doctors—factors which were significantly positively associated with taking antibiotics for COVID-19. The link between patients asking or expecting antibiotics and practitioners’ prescribing antibiotics is consistent with findings from previous studies [37,38,39]. It highlights the key role doctors play in facilitating the inappropriate use of antibiotics, especially in Australia where a doctor’s prescription is the only way for dispensing antibiotics. Given participants reported high level of trust in doctors regarding whether they need antibiotics, it is considered that doctors play an influential role in educating their patients about the function and appropriate use of antibiotics.

These findings provide further insight into how education and knowledge building can be used as a method for managing the misuse of antibiotics. We observed that antibiotic takers (compared to non-takers) were more highly educated. Furthermore, they were more likely to be trained in a health-related discipline. Interestingly, we also found that more antibiotic takers reported that they knew the difference between a viral versus bacterial infection despite holding lower levels of knowledge about the correct function of antibiotics in curing bacterial infections, not the flu. These results suggest that educational campaigns may be enhanced by delivering key messages about the precise function of antibiotics and in ways that appeal to more highly educated individuals in the population.

This study extends our understanding of the extent to which self-medication with antibiotics occurs, by revealing that some people will self-medicate with antibiotics for preventative purposes. Research so far has mainly focused on self-medication with antibiotics for treating self-diagnosed illness or symptoms. Our study reveals that self-medication with antibiotics may also be used for the prevention of disease, as reflected by almost 20% of our sample taking antibiotics to avoid being infected with COVID-19. This preventative behaviour was significantly associated with psychological distress related to COVID-19. While the current study supports the current understanding that antibiotic takers may lack knowledge of the function of antibiotics (e.g., that antibiotics can treat the flu), our findings extend this further by suggesting that psychological responses such as distressing feelings and emotions also play a role in self-medication with antibiotics. These findings have significant implications for developing effective strategies to manage misuse of antibiotics. For example, beyond the usual education-based strategies, our results suggest that it is important to allay patients’ anxiety, worry and helplessness regarding perceived health risks. This could be achieved by proposing alternative strategies to enhance one’s sense of autonomy, control, and positive thinking, for example.

While the present study has extended our understanding of the practice of self-medicating with antibiotics for preventive purpose by highlighting an additional underlying psychological mechanism, several limitations should also be noted. As COVID-19 is an unprecedented and global health crisis that can affect any individual with potentially fatal consequences, the findings may not generalized to more ordinary situations (e.g., taking antibiotics for protection in flu season). Future research should therefore examine whether stressful emotional reactions to potential health risks occurs in these situations. It would also be interesting to explore whether health-related distress is an enduring personal trait rather than a transient state-based feeling. Insights into these aspects would help in the development of more effective strategies aimed at combating the inappropriate use of antibiotics. In addition, as Australia is a developed country with strict regulations and antimicrobial stewardship in place, the findings may not be applicable to developing countries that lack such regulations and controls. Finally, limitations in study design and sampling need to be noted. Although we adapted a commonly used measure for knowledge of antibiotics—where participants were asked whether antibiotics can cure a number of illness including flu or common cold (yes, or no), the answers may not truly reflect the breadth of what participants really know. Similarly, although the present study did not find a link between taking antibiotics for COVID-19 and awareness of AMR, we did not measure how much exactly participants knew about AMR. Future research could therefore examine the relationship between objective and subjective knowledge of antibiotics and AMR through qualitative approach. This would elucidate whether educating people about antibiotics and AMR would be a fruitful endeavour. Although the sample of the current study was age and gender representative of the general population, the participants were recruited from a commercial participant panel where people participate in research for payment with a completion rate of 79.5%. Hence, they may not be fully representative of the general population. Consequently, the findings of the present study may be biased and could not be generalized to the general population.

## 4. Materials and Methods

### 4.1. Procedure

At the height of the COVID-19 first-wave outbreak in Australia between 16 March to 1 April 2020 (see Figure 2), a professional research service provider was engaged to conduct an online survey. The survey was approved by the Human Research Ethics Committee of the Commonwealth Scientific and Industrial Research Organisation, Australia, in line with the National Statement on Ethical Conduct in Human Research (Ethics Clearance No. 017/20). The survey sample was age and gender representative of six major capital cities in Australia (Adelaide, Brisbane, Canberra, Melbourne, Perth, and Sydney). The sample for each city was proportionally drawn in line with its population size. The survey link was emailed to potential participants identified from the service provider’s participant panel. This panel comprises of members of the public who register to participate in research surveys. The age and gender representativeness of the sample who completed survey was reviewed daily for further recruiting of potential participants. Consistent with human research ethics procedures, participants were informed that no personal identifiable information would be collected, that their survey results would remain confidential, and that their participation was voluntary. After reading the information about the survey, participants were instructed to click the ‘Start’ button to proceed if they consented to take part in the survey. Participants who completed the survey were paid a small fee for their participation.

### 4.2. Measures

The survey questions were presented in two separate sections with Part A referring to antibiotics usage and Part B focussing on Coronavirus. At the end of the survey, participants were informed that the World Health Organisation (WHO) had advised that antibiotics should not be used as a means of prevention or treatment of the Coronavirus. In addition, a link to the WHO’s website and information regarding Australian government’s helpline for COVID-19 and mental health helpline were presented.

#### 4.2.1. Measures Taken for Protecting against COVID-19

Participants were asked to indicate whether they were taking (0 = No, 1 = Yes) any of the following measures to protect themselves from coronavirus: wearing a mask, washing hands after arriving home, avoiding physical contact including handshake or kiss, avoiding crowded places, reducing the number of times going out, avoiding public transport, cancelling planned parties/gatherings, and taking antibiotics.

#### 4.2.2. Psychological Distress

The Florida Shock Anxiety Scale (FSAS) [40,41] was adapted to measure the key aspects of psychological distress. The FSAS was developed to measure patients’ psychological distress caused by the threat and fear of potential implantable cardioverter defibrillators (ICD) shock. The psychological distress elicited from the anticipation of experiencing the ICD shock is very similar to that caused by the perceived threat and fear of infection of the Coronavirus, where an infection may happen but was not certain. Hence, FSAS was adapted to measure the psychological distress associated with the COVID-19 pandemic. Participants were asked to rate their feelings towards the Coronavirus using a 5-point scale (1 = not at all, 3 = moderately, 5 = very much) on the following adjectives: angry, helpless, panicked, sad, scared, and worried. Scores on the six items were averaged such that higher scores indicating higher levels of psychological distress. The internal consistency of the scale (*α*) was 0.88, indicating the items assess the same construct—psychological distress.

#### 4.2.3. Knowledge of Antibiotics

Adapted from previous research [17,42], participants were asked to indicate whether antibiotics can treat the following conditions: common cold, flu, bacterial infection, and viral infection. They were further asked how much they knew about the difference between a viral infection and a bacterial infection (1 = know nothing at all, 5 = know a lot), and whether they had heard of the term antibiotic resistance.

#### 4.2.4. Experiences of Using Antibiotics

Experiences of using antibiotics focused on the practice of antibiotic use and easy access to antibiotic medication through social networks. Adapted from previous research [17,42], the measure includes the following questions: “Have you ever taken antibiotics when you had a cold or flu in the past?”; “Have you ever stopped taking antibiotics when you started feeling better?”; “Have you ever taken antibiotics that were left over from a previous prescription or friends’ or family members’ prescription?”; and “Could you easily get antibiotics from your family/a friend/household?”. Answers to those questions were recorded as ‘yes’, ‘no’, or ‘not sure’. In addition, participants were asked how many times they had used antibiotics in the past 12 months.

#### 4.2.5. Relationship with Doctors

Relationship with doctors was measured in relation to antibiotics prescription and trust in doctor. Adapted from Byrne, Miellet [43], antibiotics prescription was measured with: “Could you easily get antibiotics from a doctor?” (‘yes’, ‘no’, or ‘not sure’) and “Have you ever asked your doctor to prescribe antibiotics for an illness?”. Trust in doctors was measured by asking participants to indicate the extent to which they agreed with the following two statements: “I trust my doctor’s decision on whether I need antibiotics” and “I follow my doctor’s instruction on how to use antibiotics” (1 = strongly disagree, 5 = strongly agree).

#### 4.2.6. Demographic Characteristics

Participants’ gender, age, and education was recorded. In addition, the following questions were asked: “Are you trained in or do you work in a health-related field?” and “Are any of your family or friends health workers?”.

### 4.3. Data Analysis

To examine the correlates of self-medication with antibiotics for protection against COVID-19, multivariate logistic regression analysis was conducted to test the significance of each of the variables in predicting antibiotic taking for protection against COVID-19. Two-tailed independent *t*-tests for continuous variables and two-tailed Chi-squared (*χ*^2^) tests for categorical variables were conducted to examine the differences in significant variables between antibiotic takers and antibiotic non-takers.

## 5. Conclusions

Self-medication with antibiotics is widely practised globally, and is a practice which contributes to the development of antimicrobial resistance. Understanding the factors associated with inappropriate usage of antibiotics can help identify pathways and opportunities for intervention. However, the role of psychological distress and usage for the purpose of prevention over health risk was less understood. Build on existing literature, the current study sought to examine the psychological distress associated with the prevailing threat of illness and taking antibiotics in the context of the COVID-19 pandemic. In our survey with members of the Australian population, around 1 in 5 people disclosed that they were self-medicating with antibiotics to protect themselves from contracting COVID-19. Such behaviour was positively associated with psychological distress associated with COVID-19, which (especially in terms of feeling panicked and scared) raised the odds of taking antibiotics by 35%. Taking antibiotics preventatively was significantly associated with a lack of knowledge about the correct therapeutic application of antibiotics (i.e., to treat bacterial infections, not viral infections), previous inappropriate use of antibiotics (such as taking leftover antibiotics and acquiring antibiotics from friends/family), and certain aspects of the doctor-patient relationship (in terms of being more likely to ask doctors for antibiotics).

Overall, our findings suggest that to effectively manage self-medication with antibiotics, interventions may be improved by paying closer attention to emotional and psychological considerations. By allaying people’s health anxiety and building a sense of personal control and optimism, it is possible that they will be less inclined to self-medicate with antibiotics. Other effective interventions could involve interrupting established (habitual) patterns of medicating oneself by focusing on the doctor-patient interaction (i.e., prescribing practices and appropriate medicine-taking instructions), and developing tailored education programs that bust certain myths that high-risk groups seem to hold.

## Figures and Tables

**Figure 1 antibiotics-10-00232-f001:**
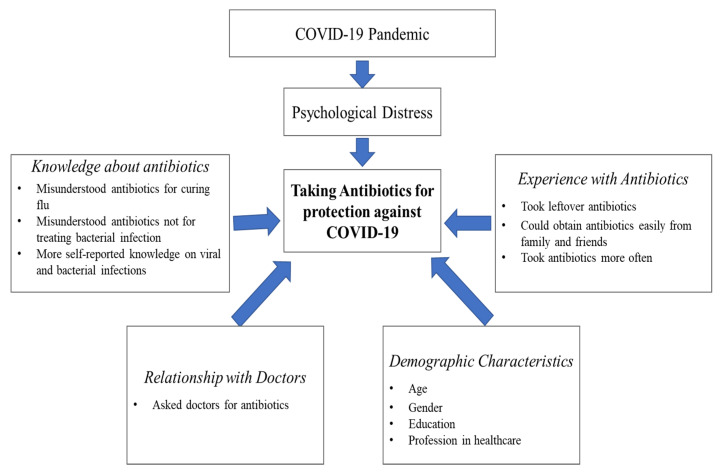
An integrative framework explaining self-medication with antibiotics for prevention against COVID-19.

**Figure 2 antibiotics-10-00232-f002:**
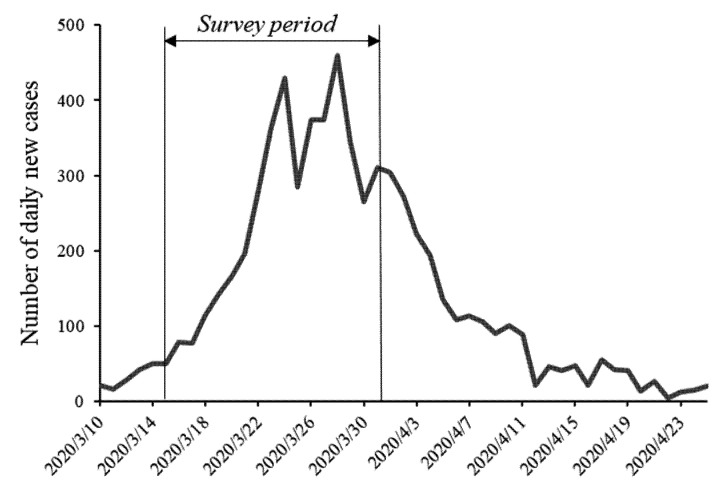
Number of daily new confirmed coronavirus cases in Australia. Data source: https://www.health.gov.au/news/health-alerts/novel-coronavirus-2019-ncov-health-alert/coronavirus-covid-19-current-situation-and-case-numbers (accessed on 1 August 2020).

**Table 1 antibiotics-10-00232-t001:** Demographic characteristics (*n* = 2217).

Demographic Variables	*n*	%
Gender		
Male	1112	50.2%
Female	1105	49.8%
Age		
18–24 years	275	12.4%
25–34 years	443	20.0%
35–44 years	435	19.6%
45–54 years	373	16.8%
55–64 years	324	14.6%
65 and over years	367	16.6%
Education		
School education (Year 10 or below)	246	11.1%
School education (Year 12)	365	16.5%
Certificate	356	16.1%
Advanced diploma/diploma	261	11.8%
Bachelor degree	632	28.5%
Graduate diploma/Graduate certificate	108	4.9%
Postgraduate	249	11.2%

**Table 2 antibiotics-10-00232-t002:** Percentage of participants taking various measures against COVID-19 (*n* = 2217).

Measures	% of Participants Taking the Measure		*χ*^2^ Test
	Antibiotic Takers	Non-Takers	
Wear a mask	63.2%	18.2%	358.64 ***
Wash hands after arriving home	92.4%	89.7%	2.41, ns
Avoid physical contact including handshake or kiss	90.3%	85.7%	6.38 *
Avoid crowded places	88.9%	80.4%	16.86 ***
Reduce the number of times going out	88.4%	79.0%	19.98 ***
Avoid public transport	83.8%	70.0%	33.23 ***
Cancel planned parties/gatherings	85.2%	69.9%	41.04 ***

Note: Two-tailed *χ*^2^ test: *** *p* < 0.001, * *p* < 0.05. ns: Not significant.

**Table 3 antibiotics-10-00232-t003:** Descriptive statistics, and Odds Ratio from logistic regression analysis of predictors on taking antibiotics for protection against COVID-19 (*n* = 2217).

Predicting Variables	Descriptive Statistic	Logistic Regression
	*n* (%)/M (SD)	Odds Ratio (95% CI)
**Demographic characteristics**		
Age	3.51 (1.64)	0.89 * (0.81, 0.97)
Gender: Male vs. Female (0)	50.2% male	1.29 * (1.00, 1.67)
Education	3.90 (1.82)	1.13 ** (1.05, 1.21)
Profession: Health vs. Non-heath (0)	14.0% health; 86.0% non-health	1.63 ** (1.15, 2.32)
Family member profession: Health vs. Non-heath (0)	23.6% health; 76.4% non-health	1.13, ns (0.83, 1.53)
**Psychological distress**	2.96 (1.01)	1.35 *** (1.19, 1.54)
**Knowledge of antibiotics**		
Cure common cold: Yes vs. No (0)	18.9% yes	1.28, ns (0.94, 1.75)
Cure flu: Yes vs. No (0)	34.1% yes	1.60 *** (1.20, 2.13)
Cure bacterial infection: No vs. Yes (0)	25.6% no	1.99 *** (1.51, 2.62)
Cure viral infection: Yes vs. No (0)	28.1% yes	0.96, ns (0.72, 1.30)
Subjective knowledge on viral and bacterial infection	2.94 (1.03)	1.16 * (1.02, 1.32)
Heard about antibiotic resistance: Yes vs. No (0)	74.4% yes	0.94, ns (0.70, 1.27)
**Experiences in using antibiotics**		
Took antibiotics for cold/flu: Yes vs. No (0)	35.6% yes	1.35, ns (0.99, 1.86)
Took antibiotics for cold/flu: Unsure vs. No (0)	8.5% unsure	0.69, ns (0.40, 1.19)
Stopped taking: Yes vs. No (0)	35.1% yes	0.79, ns (0.58, 1.08)
Stopped taking: Unsure vs. No (0)	5.3% unsure	1.39, ns (0.78, 2.47)
Took leftover antibiotics: Yes vs. No (0)	23.2% yes	1.85 *** (1.36, 2.52)
Took leftover antibiotics: Unsure vs. No (0)	4.1% unsure	1.87 * (1.02, 3.43)
Easily acquire antibiotics from friends/family: Yes vs. No (0)	17.9% yes	2.03 *** (1.46, 2.82)
Easily acquire antibiotics from friends/family: Unsure vs. No (0)	7.9% unsure	1.87 * (1.12, 3.11)
Number of times taking antibiotics in the past year	1.93 (1.09)	1.24 *** (1.10, 1.39)
**Relationship with doctors**		
Easily acquire from doctor: Yes vs. No (0)	47.1% yes	0.76, ns (0.56, 1.03)
Easily acquire from doctor: Unsure vs. No (0)	15.3% unsure	0.53 ** (0.33, 0.84)
Asked doctor for antibiotics: Yes vs. No (0)	32.2% yes	1.36 * (1.00, 1.84)
Asked doctor for antibiotics: Unsure vs. No (0)	7% unsure	1.93 * (1.15, 3.24)
Trust doctor’s decision on need for antibiotics	3.96 (1.16)	0.93, ns (0.81, 1.07)
Follow doctor’s instruction on using antibiotics	4.18 (1.03)	0.90, ns (0.77, 1.05)

Note: *** *p* ≤ 0.001, ** *p* < 0.01, * *p* ≤ 0.05. ns: Not significant. M = Mean, SD = Standard Deviation. Age was coded as: 1 = 18–24 years, 2 = 25–34 years, 3 = 35–44 years, 4 = 45–54 years, 5 = 55–64 years, 6 = 65 or older. Education was coded as: 1 = Year 10 or below, 2 = completed Year 12, 3 = Certificate, 4 = Advanced diploma/diploma, 5 = Bachelor degree, 6 = Graduate diploma/Graduate certificate, 7 = Postgraduate. “Knowledge on viral and bacterial infection”, “Trust doctor’s decision on need for antibiotics”, and “Follow doctor’s instruction on using antibiotics” were measured on 5-point scales.

**Table 4 antibiotics-10-00232-t004:** Emotional reactions over COVID-19 pandemic by antibiotic takers and the non-takers.

Emotions	Antibiotic Takers	Non-Antibiotic Takers	*t*-test (95% CI of the Difference)
	M (SD)	M (SD)	
Angry	3.28 (1.28)	2.79 (1.35)	7.07 *** (0.36, 0.63)
Helpless	3.39 (1.22)	2.88 (1.26)	7.73 *** (0.38, 0.64)
Panicked	3.26 (1.26)	2.30 (1.19)	14.43 *** (0.83, 1.09)
Sad	3.42 (1.25)	3.09 (1.30)	4.85 *** (0.20, 0.46)
Scared	3.45 (1.28)	2.79 (1.24)	9.77 *** (0.53, 0.80)
Worried	3.64 (1.18)	3.28 (1.15)	5.73 *** (0.24, 0.49)

Note. 2-tailed *t*-test. *** *p* < 0.001. M = Mean, SD = Standard Deviation. All emotions are measured on a 5-point Likert scale (1 = not at all, 3 = moderately, 5 = very much).

**Table 5 antibiotics-10-00232-t005:** Frequency of taking antibiotics by both groups in the past 12 months.

	Number of Times Taking Antibiotics				
	Never	Once	Twice	Three Times	More than 3 Times
Antibiotics takers	23.4%	31.3%	27.8%	8.6%	9.0%
The non-takers	50.9%	29.0%	13.4%	3.6%	3.1%

## Data Availability

Data available on request due to privacy or ethical requirement.

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
