# Peer review of "Self-Medication with Antibiotics for Protection against COVID-19: The Role of Psychological Distress, Knowledge of, and Experiences with Antibiotics"

_antibiotics, 2021, doi:10.3390/antibiotics10030232_

Round 1
Reviewer 1 Report
This study investigates the relationship between psychological distress caused by COVID-19 and self-medication with antibiotics. Overall, the manuscript has been well written however, there are some areas that need clarity. My comments and questions are as follows.
- Author says that the survey link was emailed to potential participants identified from the service provider’s participant panel. How was the potential participants identified? What does the service provider’s participant panel means?
- How populations were sampled for six cities? Did author calculate the response rates for individual six cities? How representative were the samples from different cities?
- Comparison between antibiotic takers and antibiotic non-takers for all the outcome measures seems to be tricky. 19.5% vs 80.5%. How representatives the group samples are? It is unclear whether author took any statistical measures to avoid bias regarding this comparison for all the outcome of interest. How generalisable the findings are?
- It is not clear whether survey questionnaires were validated (e.g., Content validity, face validity etc)? There is no result in this regard as well.
- Psychological distress in many ways (e.g., fear, perceived risk) is associated with self medication of antibiotics irrespective of COVID cases. How it is different for COVID issues can be described. It would also be worthwhile to discuss the increased antibiotic use data if reported for the period of COVID and survey period to realise the impact of psychological distress
- In the tables, p-value and CI can be presented. Additionally, the number of participants should be clearly mentioned in the tables. The only percentage is tricky to interpret.
- Methodological limitations can be mentioned
- Conclusion can be more specific and to the point
Author Response
Reviewer 1 Comments and Suggestions for Authors
This study investigates the relationship between psychological distress caused by COVID-19 and self-medication with antibiotics. Overall, the manuscript has been well written however, there are some areas that need clarity. My comments and questions are as follows.
- Author says that the survey link was emailed to potential participants identified from the service provider’s participant panel. How was the potential participants identified? What does the service provider’s participant panel means?
Responses:
We have added the following to 4.1 Procedure:
“The sample for each city was proportionally drawn in line with its population size. The survey link was emailed to potential participants identified from the service provider’s participant panel. This panel comprises members of the public who register to participate in research surveys. The age and gender representativeness of the sample who completed survey was reviewed daily for further recruiting of potential participants.”
- How populations were sampled for six cities? Did author calculate the response rates for individual six cities? How representative were the samples from different cities?
Responses:
The sampling procedure is described at 4.1 Procedure.
The response rate was calculated for the whole sample as reported at the beginning of 2. Results. We did not calculate response rates for the individual six cities as this is not considered standard practice in scientific reporting.
The participants from the six capital cities were age and gender representative of each city. This has been included in 4.1 Procedure.
- Comparison between antibiotic takers and antibiotic non-takers for all the outcome measures seems to be tricky. 19.5% vs 80.5%. How representatives the group samples are? It is unclear whether author took any statistical measures to avoid bias regarding this comparison for all the outcome of interest. How generalisable the findings are?
Responses:
As per our response to the previous comment, the overall sample and the sample in each city, was representative on age and gender. The sub-groups of antibiotic takers and non-takers however differed significantly on age and gender. That is, and as shown in Table 3, there was a significantly higher proportion of males and younger people among those who took antibiotics compared to those who did not take antibiotics.
Because the analysis included these demographic variables, along with other variables (education, profession), the effects of these demographic variables are therefore statistically controlled or partialled out of the analysis.
As the unequal sample sizes of the ‘antibiotic taker’ (19.5%) and the ‘antibiotic non-taker’ (80.5%), we used the statistic results of “Equal variances not assumed”. To make it clear, we have added the following to Section 2.1.:
“Due to the unequal sample sizes of the two groups, the comparison analysis was conducted with equal variances not assumed.”
Regarding the generalisability of the findings, we have added the following sentence to the discussion:
“In addition, as Australia is a developed country with strict regulations and antimicrobial stewardship in place, the findings may not be applicable to developing countries where inappropriate use of antibiotics is widespread.”
- It is not clear whether survey questionnaires were validated (e.g., Content validity, face validity etc)? There is no result in this regard as well.
Responses:
The measures used in our study were adapted from established measures used in previous research on antibiotics use/misuse. The sources of the measures are presented in Section 4.2 Measures on Page 15-17.
The study only has one composite scale – psychological distress. Following the convention of reporting adapted measures, we have reported the internal consistency of the scale (α) to indicate the reliability of the measure.
- Psychological distress in many ways (e.g., fear, perceived risk) is associated with self medication of antibiotics irrespective of COVID cases. How it is different for COVID issues can be described. It would also be worthwhile to discuss the increased antibiotic use data if reported for the period of COVID and survey period to realise the impact of psychological distress
Responses:
Yes, we agree that psychological distress may be associated with self-medication of antibiotics irrespective of the COVID-19 pandemic. However, our study focussed squarely on this relationship in the context of the COVID-19 pandemic. We therefore summarise how the COVID-19 pandemic has induced psychological distress worldwide in section 1.2.
Investigating whether there is increased antibiotic use over time will be very informative. However, it is beyond the scope of our study.
- In the tables, p-value and CI can be presented. Additionally, the number of participants should be clearly mentioned in the tables. The only percentage is tricky to interpret.
Responses:
We have added the number of participants to all tables.
As most of the tables already contain a large amount information, adding the exact p-value will make the tables crowded. We believe using the APA standard to report p value with * is more straightforward and still provides the necessary information to readers.
We have added CIs to the tables now.
- Methodological limitations can be mentioned.
Responses:
We have added the following on methodological limitation:
“In addition, as Australia is a developed country with strict regulations and antimicrobial stewardship in place, the findings may not be applicable to developing countries that lack such regulations and controls. Finally, limitations in methodology need to be noted. Although we adapted a commonly used measure for knowledge of antibiotics – where participants were asked whether antibiotics can cure a number of illness including flu or common cold (yes, or no), the answers may not truly reflect the breadth of what participants really know. Similarly, although the present study did not find a link between taking antibiotics for COVID-19 and awareness of AMR, we did not measure how much exactly participants knew about AMR. Future research could therefore examine the relationship between objective and subjective knowledge of antibiotics and AMR, and taking antibiotics for preventive purposes. This would elucidate whether educating people about AMR would be a fruitful endeavour.”
- Conclusion can be more specific and to the point
Responses:
We have now edited the Conclusion paragraphs and endeavoured to be more specific and to the point (while recognising that another reviewer requested more discussion of the findings).
Reviewer 2 Report
Self-Medication with Antibiotics for Protection against COVID-19: The role of Psychological Distress, Knowledge of, and Experiences with Antibiotics
The abstract is descriptive and qualitative. Normally an abstract should state briefly the purpose of the study undertaken and meaningful conclusions based on the obtained results.
Hence, this needs rewriting. I would expect brief, yet concise, the quantitative data description of the results in the abstract.
The given list of keywords is superficial with broader terms. More specific terms should be used. Replace accordingly.
The introduction is short. More literature should be added with recent and relevant literature.
The novelty of the study should be clearly highlighted in the manuscript at the end of the introduction section, as there are some existing literature reports.
The conclusion is superficial. Herein, I would like to see the major findings and how they are addressing the left behind research gaps and covering current challenges.
Referencing is not right. Literature needs to be updated with care. At least 20% references should be from recent years 2017-2020.
Editorial issues: The Latin names and Greek letters should be presented in italic in the whole manuscript, the unit presentation should be unified in the whole manuscript, abbreviations presentation should be unified.
Author Response
Reviewer 2
Comments and Suggestions for Authors
Self-Medication with Antibiotics for Protection against COVID-19: The role of Psychological Distress, Knowledge of, and Experiences with Antibiotics
The abstract is descriptive and qualitative. Normally an abstract should state briefly the purpose of the study undertaken and meaningful conclusions based on the obtained results.
Hence, this needs rewriting. I would expect brief, yet concise, the quantitative data description of the results in the abstract.
Responses:
We have now rewritten the Abstract to make the context, the purpose and methodology of the study, and the key findings and implications clearer.
However, we don’t think Abstract is a place to report detailed quantitative data. Instead, it is about highlighting the meaning of the statistical data.
The given list of keywords is superficial with broader terms. More specific terms should be used. Replace accordingly.
Responses:
We have made changes to the keywords:
“Knowledge of antibiotics; antimicrobial resistance; antibiotic use for prevention; perceived health risk; Coronavirus pandemic; psychological distress”
The introduction is short. More literature should be added with recent and relevant literature.
Responses:
The Introduction was condensed in our previous revision in line with the Academic Editor’s suggestion, with a focus on presenting recent and relevant literature.
As the paper is nearly 7000 words in total already, we agree with the Academic Editor’s initial comment that a condensed Introduction is a better way to go, where the focus is on the background and the significance of the present study, while the recent and relevant literature is thoroughly but briefly covered.
The novelty of the study should be clearly highlighted in the manuscript at the end of the introduction section, as there are some existing literature reports.
Responses:
To our knowledge, there is no study so far on investigating self-medication with antibiotic for protection against from COVID-19. While we have summarised the COVID-19 associated psychological distress published recently in the Introduction already, we have not highlighted the novelty of our study more clearly in “1.4. The present study”:
“While it has been demonstrated that COVID-19 has caused significant impact on mental health among the public [e.g. 23, 24, 25], what measures people take to manage such psychological distress is less understood. Examining this research question in the context of inappropriate use of antibiotics and AMR, we aimed to investigate the role of psychological distress in self-medication with antibiotics. Hence, this study explored whether the psychological distress induced by the COVID-19 pandemic has promulgated use of antibiotics as a perceived preventative measure against contracting COVID-19.”
The conclusion is superficial. Herein, I would like to see the major findings and how they are addressing the left behind research gaps and covering current challenges.
Responses:
We have now edited the Conclusion paragraphs and endeavoured to be more detailed in our explanation of the main findings (while recognising that another reviewer requested that the Conclusion be more specific and to the point).
Referencing is not right. Literature needs to be updated with care. At least 20% references should be from recent years 2017-2020.
Response:
We are a little bit confused with this comment. Of the total 43 papers er referred in our paper, 35 are published in recent years from 2017 to 2020, which is 81% of our references. This is far more than the 20% recommended by Reviewer 2.
Editorial issues: The Latin names and Greek letters should be presented in italic in the whole manuscript, the unit presentation should be unified in the whole manuscript, abbreviations presentation should be unified.
Responses:
We have now gone through the manuscripts and made the corresponding changes as suggested by the Reviewer.
Round 2
Reviewer 1 Report
This is a much improved version.
Author Response
Thanks.